# Kernel Truncated Randomized Ridge Regression: Optimal Rates and Low Noise Acceleration

**Kwang-Sung Jun**
The University of Arizona*
kjun@cs.arizona.edu

**Ashok Cutkosky**
Google Research
ashok@cutkosky.com

**Francesco Orabona**
Boston University
francesco@orabona.com

## Abstract

In this paper, we consider the nonparametric least square regression in a Reproducing Kernel Hilbert Space (RKHS). We propose a new randomized algorithm that has optimal generalization error bounds with respect to the square loss, closing a long-standing gap between upper and lower bounds. Moreover, we show that our algorithm has faster finite-time and asymptotic rates on problems where the Bayes risk with respect to the square loss is small. We state our results using standard tools from the theory of least square regression in RKHSs, namely, the decay of the eigenvalues of the associated integral operator and the complexity of the optimal predictor measured through the integral operator.

## 1 Introduction

Given a training set $S = \{\boldsymbol{x}_t, y_t\}_{t=1}^n$ of $n$ samples drawn identically and independently distributed from a fixed but unknown distribution $\rho$ on $\mathbb{X} \times \mathbb{Y}$, the goal of nonparametric least square regression is to find a function $\hat{f}$ whose risk

$$\mathcal{R}(\hat{f}) := \int_{\mathbb{X} \times \mathbb{Y}} \left( \hat{f}(\boldsymbol{x}) - y \right)^2 d\rho$$

is close to the optimal risk

$$\mathcal{R}^\star := \inf_f \mathcal{R}(f) .$$

We focus on the kernel-based methods, which consider candidate functions from a Reproducing Kernel Hilbert Space (RKHS) of functions and possibly their composition with elementary functions.

A classic kernel-based algorithm for nonparametric least squares is Kernel Ridge Regression (KRR), which constructs the prediction function $\hat{f}$ as

$$\hat{f} = \underset{f \in \mathcal{H}_K}{\operatorname{argmin}} \ \lambda \|f\|^2 + \frac{1}{n} \sum_{t=1}^n (f(\boldsymbol{x}_t) - y_t)^2,$$

where $\mathcal{H}_K$ is a RKHS associated with a kernel $K$ and $\lambda$ is the hyperparameter controlling the amount of regularization.

It has been proved that, when the amount of regularization is chosen optimally and under similar assumptions, KRR converges to the Bayes risk at the best known rate among kernel-based algorithms [Lin et al., 2018]. Despite this result, kernel-based learning is still not a solved problem: these rates match the known lower bounds in Fischer and Steinwart [2017] only in some regimes, unless additional assumptions are used [Steinwart et al., 2009]. Indeed, it was not even known if the lower bound was optimal in all the regimes [Pillaud-Vivien et al., 2018].

Moreover, recent empirical results have also challenged the theoretical results. In particular, KRR without regularization seems to perform very well on real-world datasets [Zhang et al., 2017, Belkin et al., 2018], at least in the classification setting, and even outperform KRRs with any nonzero regularization in a popular computer vision dataset [Liang and Rakhlin, 2018, Figure 1]. This challenges the theoretical findings because our current understanding of kernel-based learning tells us that a non-zero regularization is needed in all cases for learning in infinite dimensional RKHSs. Given the current gap in upper and lower bounds, it is unclear if this mismatch between theory and practice is due to $(i)$ suboptimal analyses that lead to suboptimal choices of the amount of regularization or $(ii)$ not taking into account crucial data-dependent quantities (e.g., capturing "easiness" of the problem) that allow fast rates and minimal regularization.

In this work, we address all these questions. We propose a new kernel-based learning algorithm named Kernel Truncated Randomized Ridge Regression (KTR$^3$). We show that the performance of KTR$^3$ is minimax optimal, matching known lower bounds. This closes the gap between upper and lower bounds, without the need for additional assumptions. Moreover, we show that the generalization guarantee of KTR$^3$ accelerates when the Bayes risk is zero or close to zero. As far as we know, the phenomenon is new in this literature. Finally, we identify a regime of easy problems in which the best amount of regularization is exactly zero.

Another important contribution lies in our proof methods, which vastly differ from the usual one in this field. In particular, we use methods from the online learning literature that make the proof very simple and rely only on population quantities rather than empirical ones. We believe the community of nonparametric kernel-regression will greatly benefit from the addition of these new tools.

The rest of the paper is organized as follows: In the next section, we formally introduce the setting and our assumptions. In Section 3 we introduce our KTR$^3$ algorithm and its theoretical guarantee, and in Section 4 the precise comparison with similar results. In Section 5, we empirically evaluate our findings. Finally, Section 6 discusses open problems and future directions of research.

## 2   Setting and Notation: Source Condition and Eigenvalue Decay

In this section, we formally introduce our learning setting and our characterization of the complexity of each regression problem. This characterization is standard in the literature on regression in RKHS, see, e.g., Steinwart and Christmann [2008], Steinwart et al. [2009], Dieuleveut and Bach [2016], Lin et al. [2018].

Let $\mathbb{X} \subset \mathbb{R}^d$ a compact set and $\mathcal{H}_K$ a separable RKHS associated to a Mercer kernel $K : \mathbb{X} \times \mathbb{X} \to \mathbb{R}$ implementing the inner product $\langle \cdot, \cdot \rangle$ and induced norm $\| \cdot \|$. The inner product is defined so that it satisfies the reproducing property, $\langle K(\boldsymbol{x}, \cdot), f(\cdot) \rangle = f(\boldsymbol{x})$. Denote by $K_t \in \mathbb{R}^{t \times t}$ the *Gram matrix* such that $K_{i,j} = K(\boldsymbol{x}_i, \boldsymbol{x}_j)$ where $\boldsymbol{x}_i, \boldsymbol{x}_j$ belong to $S_t \subseteq S$ that contains the first[2] $t$ elements of the training set $S$.

Our first assumption is related to the boundedness of the kernel and labels.

**Assumption 1** (Boundedness). *We assume $K$ to be bounded, that is, $\sup_{\boldsymbol{x} \in \mathbb{X}} K(\boldsymbol{x}, \boldsymbol{x}) = R^2 < \infty$. To avoid superfluous notations and without loss of generality, we further assume $R = 1$. We also assume the labels to be bounded: $\mathbb{Y} = [-Y, Y]$ where $Y < \infty$.*

Denote by $\rho_{\mathbb{X}}$ the marginal probability measure on $\mathbb{X}$ and let $\mathcal{L}^2_{\rho_{\mathbb{X}}}$ be the space of square integrable functions with respect to $\rho_{\mathbb{X}}$. We will assume that the support of $\rho_{\mathbb{X}}$ is $\mathbb{X}$, whose norm is denoted by $\|g\|_\rho := \sqrt{\int_{\mathbb{X}} g^2(\boldsymbol{x}) d\rho_{\mathbb{X}}}$. It is well known that the function minimizing the risk over all functions in $\mathcal{L}^2_{\rho_{\mathbb{X}}}$ is $f_\rho(\boldsymbol{x}) := \int_{\mathbb{Y}} y d\rho(y|\boldsymbol{x})$, which has the *Bayes risk* with respect to the square loss, $\mathcal{R}^\star = \mathcal{R}(f_\rho) = \inf_{f \in \mathcal{L}^2_{\rho_{\mathbb{X}}}} \mathcal{R}(f)$.

If we use a Universal Kernel (e.g., the Gaussian kernel) [Steinwart, 2001] and $\mathbb{X}$ is compact, we have that $\inf_{f \in \mathcal{H}_K} \mathcal{R}(f) = \mathcal{R}^\star$ [Steinwart and Christmann, 2008, Corollary 5.29]. This suggests that using a universal kernel is somehow enough to reach the Bayes risk. However, while $f_\rho \in \mathcal{L}^2_{\rho_{\mathbb{X}}}$, this actually does not imply that $f_\rho \in \mathcal{H}_K$ but only that $f_\rho \in \overline{\mathcal{H}_K}$, which is the closure of $\mathcal{H}_K$. Thus, the question

**Algorithm 1** KTR$^3$: Kernel Truncated Randomized Ridge Regression

---
**Input:** A training set $S = \{(\boldsymbol{x}_i, y_i)\}_{i=1}^n$, a regularization parameter $\lambda \geqslant 0$
Randomly permute the training set $S$
**for** $t = 0, 1, \ldots, n-1$ **do**
&emsp;&emsp;Set $f_t = \operatorname{argmin}_{f \in \mathcal{H}_K} \lambda \|f\|^2 + \frac{1}{n} \sum_{i=1}^t (f(\boldsymbol{x}_i) - y_i)^2$
&emsp;&emsp;(take the minimum norm solution when there is no unique solution)
**end for**
Return $T^Y \circ f_k$, where $k$ is uniformly at random between 0 and $n-1$

---

of whether it is possible to achieve the Bayes risk is relevant even for Universal kernels. We address this by the standard parametrization called *source condition* that smoothly characterizes whether $f_\rho$ belongs or not to $\mathcal{H}_K$. To introduce the formalism, let $L_K : \mathcal{L}_{\rho_{\mathbb{X}}}^2 \to \mathcal{L}_{\rho_{\mathbb{X}}}^2$ be the integral operator defined by $(L_K f)(\boldsymbol{x}) = \int_{\mathbb{X}} K(\boldsymbol{x}, \boldsymbol{x}') f(\boldsymbol{x}') d\rho_{\mathbb{X}}(\boldsymbol{x}')$. There exists an orthonormal basis $\{\Phi_1, \Phi_2, \cdots\}$ of $\mathcal{L}_{\rho_{\mathbb{X}}}^2$ consisting of eigenfunctions of $L_K$ with corresponding non-negative eigenvalues $\{\lambda_1, \lambda_2, \cdots\}$ and the set $\{\lambda_i\}$ is finite or $\lambda_k \to 0$ when $k \to \infty$ [Cucker and Zhou, 2007, Theorem 4.7]. Since $K$ is a Mercer kernel, $L_K$ is compact and positive. Moreover, given that we assumed the kernel to be bounded, $L_K$ is trace class, hence compact [Steinwart and Christmann, 2008]. Therefore, the fractional power operator $L_K^\beta$ is well-defined for any $\beta \geqslant 0$. We indicate its range space by

$$L_K^\beta(\mathcal{L}_{\rho_{\mathbb{X}}}^2) := \left\{ f = \sum_{i=1}^\infty \lambda_i^\beta a_i \Phi_i \; : \; \sum_{i=1}^\infty a_i^2 < \infty \right\}.$$

This space has a key role in our analysis. In particular, we will use the following assumption.

**Assumption 2** (Source Condition). *We assume that* $f_\rho \in L_K^\beta(\mathcal{L}_{\rho_{\mathbb{X}}}^2)$ *for* $0 < \beta \leqslant \frac{1}{2}$, *which is*

$$\exists g \in \mathcal{L}_{\rho_{\mathbb{X}}}^2 \; : \; f_\rho = L_K^\beta(g).$$

Note that the assumption above is always satisfied for $\beta = 0$ because, by definition of the orthonormal basis, $L_K^0(\mathcal{L}_{\rho_{\mathbb{X}}}^2) = \mathcal{L}_{\rho_{\mathbb{X}}}^2$. On the other hand, we have that $L_K^{1/2}(\mathcal{L}_{\rho_{\mathbb{X}}}^2) = \mathcal{H}_K$, that is every function $f \in \mathcal{H}_K$ can be written as $L_K^{1/2} g$ for some $g \in \mathcal{L}_{\rho_{\mathbb{X}}}^2$, and $\|f\| = \|L_K^{-1/2} f\|_\rho$ [Cucker and Zhou, 2007, Corollary 4.13]. Hence, *the values of $\beta$ in $[0, \frac{1}{2}]$ allow us to consider spaces in between $\mathcal{L}_{\rho_{\mathbb{X}}}^2$ and $\mathcal{H}_K$,* including the extremes. Thus, a bigger $\beta$ means a simpler function $f_\rho$.

Another assumption needed to characterize the learning process is on the *complexity of the RKHS itself*, rather than on the complexity of the optimal function. This is typically done assuming that the eigenvalue of the integral operator satisfies a certain rate of decay. We will use equivalent condition, assuming that the trace of some fractional power of the integral operator is bounded.

**Assumption 3** (Eigenvalue Decay). *Assume that there exists $b \in [0, 1]$ such that* $\operatorname{Tr}[L_K^b] < \infty$.

Note that the sum of the eigenvalues of $L_K$ is at most $\sup_{\boldsymbol{x} \in \mathbb{X}} K(\boldsymbol{x}, \boldsymbol{x})$, which we assumed to be bounded in Assumption 1. This implies that the assumption above is always satisfied with $b = 1$. Hence, a smaller $b$ corresponds to an RKHS with a smaller complexity.

## 3 Kernel Truncated Randomized Ridge Regression

We now describe our algorithm called Kernel Truncated Randomized Ridge Regression (KTR$^3$). The pseudo-code is in Algorithm 1. The algorithm consists of two stages. In the first stage, we generate $n$ candidate functions solving KRR with increasing sizes of the training set and a fixed regularization weight $\lambda$. In the second stage, we select the prediction function as the truncation of one of the candidate functions uniformly at random. Note that this is equivalent to extracting a subset of the training set of size $i$, where $i$ is uniformly at random between 0 and $n-1$ and training a KRR on the subset with parameter $\lambda$. The truncation function is defined as follows

$$T^Y(z) := \min(Y, |z|) \cdot \operatorname{sign}(z).$$

The definition of the truncation function implies that $(T^Y(\hat{y}) - y)^2 \leqslant (\hat{y} - y)^2, \forall \hat{y} \in \mathbb{R}, y \in \mathbb{Y}$.

We now present our two main theorems on the excess risk of KTR[3] where Theorem 1 is on $\lambda > 0$ and Theorem 2 is on $\lambda = 0$ for an "easy" problem regime. The proof of Theorem 2 is in the Appendix.

**Theorem 1.** *Let $\mathbb{X} \subset \mathbb{R}^d$ be a compact domain and $K$ a Mercer kernel such that Assumptions 1,2, and 3 are verified. Define by $f_{S,\lambda}$ the function returned by the KTR[3] algorithm on a training set $S$ with regularization parameter $\lambda > 0$. Then*

$$\mathbb{E}\left[\mathcal{R}(f_{S,\lambda})\right] - \mathcal{R}(f_\rho)$$
$$\leqslant \lambda^{2\beta}\|L_K^{-\beta}f_\rho\|_\rho^2 + \min\left[\frac{4Y^2\operatorname{Tr}[L_K^b]}{\lambda^b n}\min\left(\ln^{1-b}\left(1 + \frac{1}{\lambda}\right), \frac{1}{b}\right), \frac{\lambda^{2\beta-1}\|L_K^{-\beta}f_\rho\|_\rho^2}{n} + \frac{\mathcal{R}(f_\rho)}{\lambda n}\right],$$

*where the expectation is with respect to $S$ and the randomization of the algorithm.*

**Theorem 2.** *Let $\lambda = 0$ and assume the same conditions as in Theorem 1 except for $\lambda$. Assume $\beta = 1/2$ and $\mathcal{R}(f_\rho) = 0$. Assume that the distribution $\rho$ satisfies that $K_n$ is invertible with probability 1. Then, $\mathbb{E}\left[\mathcal{R}(f_{S,0})\right] - \mathcal{R}(f_\rho) = O(n^{-1})$.*

**Remark.** Our algorithm can be changed to randomize at the prediction time for each test data point rather than the training time while enjoying the same risk bound. Furthermore, our algorithm can sample from $\lfloor(1-\alpha)n\rfloor$ to $n-1$ for some $\alpha \in (0,1]$ instead of from $\{0, \ldots, n-1\}$ and obtain a rate $\frac{1}{\alpha}$ factor worse than the bounds above; our choice of presentation of Algorithm 1 is for simplicity.

From the above theorem, with appropriate settings of the regularization parameter $\lambda$ it is possible to obtain the following convergence rates.

**Corollary 1.** *Under the assumptions of Theorem 1, there exists a setting of $\lambda \geqslant 0$ such that:*

*(i) When $b \neq 0$,*

$$\mathbb{E}\left[\mathcal{R}(f_{S,\lambda})\right] - \mathcal{R}(f_\rho) \leqslant O\left(\min\left((n/\mathcal{R}(f_\rho))^{-\frac{2\beta}{2\beta+1}} + n^{-2\beta}, n^{-\frac{2\beta}{2\beta+b}}\right)\right).$$

*(ii) In the case $b = 0$ and $\beta = \frac{1}{2}$,[3]*

$$\mathbb{E}\left[\mathcal{R}(f_{S,\lambda})\right] - \mathcal{R}(f_\rho) \leqslant O\left(n^{-1}\operatorname{Tr}[L_K^0]\log\left(1 + n/\operatorname{Tr}[L_K^0]\right)\right).$$

The proof and the tuning of $\lambda$ can be found in the Appendix. Before moving to the proof of Theorem 1 in the next section, there are some interesting points to stress.

- In the case of $\mathcal{R}(f_\rho) \neq 0$, our rate $n^{-\frac{2\beta}{2\beta+b}}$ matches the worst-case lower bound [Fischer and Steinwart, 2017] without additional assumptions for the first time in the literature, to our knowledge. Specifically, our bound is a strict improvement in the regime $2\beta + b < 1$ upon the best-known bound $O(n^{-2\beta})$ of KRR [Lin et al., 2018] and stochastic gradient descent [Dieuleveut and Bach, 2016]. In this regime, our rate goes to $O(n^{-1})$ as $b$ goes to 0.

- If $\mathcal{R}(f_\rho) = 0$, we have convergence of the risk to 0 at a faster rate of $n^{-\frac{2\beta}{\min\{2\beta+b,1\}}}$. It is important to stress that this holds also in the case that $f_\rho \notin \mathcal{H}_K$, i.e., $\beta < \frac{1}{2}$. As far as we know, this result is new and we are not aware of lower bounds under the same assumptions.

- When $\mathcal{R}(f_\rho) = 0$, the optimal $\lambda$ that minimizes the generalization upper bound in Theorem 1 goes to zero when $\beta$ goes to $1/2$ and becomes exactly 0 when $\beta$ is exactly $1/2$.

### 3.1  Proof of Theorem 1

Our proof technique is vastly different from the existing ones for analyzing KRR and stochastic gradient descent methods. It is also extremely short and simple compared to the proofs of similar results. Our technique is based on the well-known possibility to solve batch problems through a reduction to online learning ones. In turn, we use a recent result on the performance of online kernel ridge regression, Theorem 3 by Zhdanov and Kalnishkan [2013]. This result is the key to obtain the improved rates in the regime $2\beta + b < 1$. In particular, it allows us to analyze the effect of the

eigenvalues using only the expectation of the Gram matrix $K_n$ and nothing else. Instead, previous proofs [e.g., Lin and Cevher, 2018] involved the study of the convergence of empirical covariance operator to the population one, which seems to deteriorate when the regularization parameter becomes too small, which is precisely needed in the regime $2\beta + b < 1$.

**Theorem 3.** *[Zhdanov and Kalnishkan, 2013, Theorem 1] Take a kernel $K$ on a domain $\mathbb{X}$ and a parameter $\lambda > 0$. Then, with the notation of Algorithm 1, we have*

$$\frac{1}{n} \sum_{t=1}^{n} \frac{(f_{t-1}(\boldsymbol{x}_t) - y_t)^2}{1 + \frac{d_t}{\lambda n}} = \min_{f \in \mathcal{H}_K} \lambda \|f\|^2 + \frac{1}{n} \sum_{t=1}^{n} (f(\boldsymbol{x}_t) - y_t)^2 ,$$

*where $d_t := K(\boldsymbol{x}_t, \boldsymbol{x}_t) - \boldsymbol{k}_{t-1}(\boldsymbol{x}_t)^\top (K_{t-1} + \lambda n I)^{-1} \boldsymbol{k}_{t-1}(\boldsymbol{x}_t) \geqslant 0$, $\boldsymbol{k}_{t-1}(\boldsymbol{x}_t) := [K(\boldsymbol{x}_t, \boldsymbol{x}_1), \dots, K(\boldsymbol{x}_t, \boldsymbol{x}_{t-1})]^\top$, and $K_{t-1}$ is the Gram matrix of the samples $\boldsymbol{x}_1, \dots, \boldsymbol{x}_{t-1}$.*

We use the following well-known result to upper bound the approximation error, which is the gap between the value of the regularized population risk minimization problem and the Bayes risk.

**Theorem 4.** *[Cucker and Zhou, 2007, Proposition 8.5.ii] Let $\mathbb{X} \subset \mathbb{R}^d$ be a compact domain and $K$ a Mercer kernel such that Assumption 2 holds. Then, for any $0 < \beta \leqslant 1/2$, we have*

$$\min_{f \in \mathcal{H}_K} \lambda \|f\|^2 + \mathcal{R}(f) - \mathcal{R}(f_\rho) \leqslant \lambda^{2\beta} \|L_K^{-\beta} f_\rho\|_\rho^2 .$$

We also need the following technical lemmas. The proof of the next lemma is in the Appendix.

**Lemma 1.** *Under Assumptions 1 and 3, and with $\lambda > 0$, we have*

$$\mathbb{E}_S \left[ \ln \frac{|\lambda I_n + \frac{1}{n} K_n|}{|\lambda I_n|} \right] \leqslant \min \left( \ln^{1-b} \left( 1 + \frac{1}{\lambda} \right), \frac{1}{b} \right) \frac{\text{Tr}[L_K^b]}{\lambda^b} .$$

*Furthermore, if $b = 0$, then*

$$\mathbb{E}_S \left[ \ln \frac{|\lambda I_n + \frac{1}{n} K_n|}{|\lambda I_n|} \right] \leqslant \ln \left( 1 + \frac{1}{\text{Tr}[L_K^0]\lambda} \right) \text{Tr}[L_K^0] .$$

Note that the logarithmic term is unavoidable when $b = 0$ because in the finite dimensional case we pay $-\ln(\lambda)$ due to the online learning setting. The last lemma is a classic result in online learning [e.g. Cesa-Bianchi et al., 2005].

**Lemma 2.** *With the notation in Theorem 3, we have that*

$$\sum_{t=1}^{n} \frac{d_t}{d_t + \lambda n} \leqslant \ln \frac{|\lambda I_n + \frac{1}{n} K_n|}{|\lambda I_n|} .$$

*Proof.* From the elementary inequality $\ln(1 + x) \geqslant \frac{x}{x+1}$, we have that $\frac{d_t}{d_t + \lambda n} \leqslant \ln \left(1 + \frac{d_t}{\lambda n}\right)$. Hence, $\sum_{t=1}^{n} \frac{d_t}{d_t + \lambda n} \leqslant \log \prod_{t=1}^{n} \left(1 + \frac{d_t}{\lambda n}\right)$. Also, using Zhdanov and Kalnishkan [2013, Lemma 3] we have $\prod_{t=1}^{n} (\lambda n + d_t) = |\lambda n I_n + K_n|$. Putting all together, we have the stated bound. $\square$

We are now ready to prove Theorem 1.

*Proof of Theorem 1.* Define $f_\lambda = \text{argmin}_{f \in \mathcal{H}_K} \lambda \|f\|^2 + \mathcal{R}(f)$, which is the solution of the regularization true risk minimization problem.

First, we use the so-called online-to-batch conversion [Cesa-Bianchi et al., 2004] to have

$$\mathbb{E}_{S,k}[\mathcal{R}(T^Y \circ f_k)] = \mathbb{E}_S \left[ \frac{1}{n} \sum_{t=0}^{n-1} \mathcal{R}\left(T^Y \circ f_t\right) \right] = \mathbb{E}_S \left[ \frac{1}{n} \sum_{t=0}^{n-1} \mathbb{E}_{S_t} \mathcal{R}\left(T^Y \circ f_t\right) \right]$$

$$= \mathbb{E}_S \left[ \frac{1}{n} \sum_{t=0}^{n-1} \mathbb{E}_{S_t} \left(T^Y(f_t(\boldsymbol{x})) - y\right)^2 \right] = \mathbb{E}_S \left[ \frac{1}{n} \sum_{t=0}^{n-1} \mathbb{E}_{S_t} \left(T^Y(f_t(\boldsymbol{x}_{t+1})) - y_{t+1}\right)^2 \right]$$

$$= \mathbb{E}_S \left[ \frac{1}{n} \sum_{t=1}^{n} \left(T^Y(f_{t-1}(\boldsymbol{x}_t)) - y_t\right)^2 \right] .$$

Denote by $d'_t = \frac{d_t}{\lambda n}$, $\ell'_t = (T_Y(f_{t-1}(\boldsymbol{x}_t)) - y_t)^2$, and $\ell_t = (f_{t-1}(\boldsymbol{x}_t) - y_t)^2$. We have that

$$\mathbb{E}_S \left[ \frac{1}{n} \sum_{t=1}^n \left( T^Y(f_{t-1}(\boldsymbol{x}_t)) - y_t \right)^2 \right] = \mathbb{E}_S \left[ \frac{1}{n} \sum_{t=1}^n \ell'_t \right] = \mathbb{E}_S \left[ \frac{1}{n} \sum_{t=1}^n \frac{\ell'_t d'_t}{1 + d'_t} \right] + \mathbb{E}_S \left[ \frac{1}{n} \sum_{t=1}^n \frac{\ell'_t}{1 + d'_t} \right]$$

$$\leqslant \mathbb{E}_S \left[ \frac{1}{n} \sum_{t=1}^n \frac{\ell'_t d'_t}{1 + d'_t} \right] + \mathbb{E}_S \left[ \frac{1}{n} \sum_{t=1}^n \frac{\ell_t}{1 + d'_t} \right] .$$

We now focus on the first sum in the last inequality and we upper bound it in two different ways. First, using Lemma 2 and Lemma 1, we have

$$\mathbb{E}_S \left[ \frac{1}{n} \sum_{t=1}^n \frac{\ell'_t d'_t}{1 + d'_t} \right] \leqslant 4Y^2 \mathbb{E}_S \left[ \frac{1}{n} \sum_{t=1}^n \frac{d'_t}{1 + d'_t} \right] \leqslant \frac{4Y^2}{n} \mathbb{E}_S \left[ \ln \frac{|\lambda I + \frac{1}{n} K_n|}{|\lambda I|} \right]$$

$$\leqslant \frac{4Y^2}{n} \min \left( \ln^{1-b} \left( 1 + \frac{1}{\lambda} \right), \frac{1}{b} \right) \frac{\mathrm{Tr}[L_K^b]}{\lambda^b} .$$

Also, we can upper bound the same term as

$$\mathbb{E}_S \left[ \frac{1}{n} \sum_{t=1}^n \frac{\ell'_t d'_t}{1 + d'_t} \right] \leqslant \mathbb{E}_S \left[ \frac{1}{n} \sum_{t=1}^n \frac{\ell_t d'_t}{1 + d'_t} \right] \leqslant \mathbb{E}_S \left[ \frac{\max_t d'_t}{n} \sum_{t=1}^n \frac{\ell_t}{1 + d'_t} \right] \leqslant \frac{1}{\lambda n} \mathbb{E}_S \left[ \frac{1}{n} \sum_{t=1}^n \frac{\ell_t}{1 + d'_t} \right] .$$

Now, using Theorems 3 and 4 with the fact that $d_t \leqslant 1$, we bound the term $\mathbb{E}_S \left[ \frac{1}{n} \sum_{t=1}^n \frac{\ell_t}{1 + d'_t} \right]$ as

$$\mathbb{E}_S \left[ \frac{1}{n} \sum_{t=1}^n \frac{\ell_t}{1 + d'_t} \right] = \mathbb{E}_S \left[ \min_{\mathcal{H}_K} \lambda \|f\|^2 + \frac{1}{n} \sum_{t=1}^n (f(\boldsymbol{x}_t) - y_t)^2 \right] \leqslant \mathbb{E}_S \left[ \lambda \|f_\lambda\|^2 + \frac{1}{n} \sum_{t=1}^n (f_\lambda(\boldsymbol{x}_t) - y_t)^2 \right]$$

$$= \lambda \|f_\lambda\|^2 + \mathcal{R}(f_\lambda) = \min_{f \in \mathcal{H}_K} \lambda \|f\|^2 + \mathcal{R}(f) - \mathcal{R}(f_\rho) + \mathcal{R}(f_\rho)$$

$$\leqslant \lambda^{2\beta} \|L_K^{-\beta} f_\rho\|_\rho^2 + \mathcal{R}(f_\rho) .$$

Putting all together, we have the stated bound. $\qquad\square$

## 4 Detailed Comparison with Previous Results

The sheer volume of research on regression, see, e.g., Lin and Cevher [2018, Table 1], precludes a complete survey of the results. In this section, we focus on the closely related ones that involve infinite dimensional spaces.

First, it is useful to compare our convergence rate to the one we would get from known guarantees for KRR. We can compare it to the stability bound in Shalev-Shwartz and Ben-David [2014] for KRR:

$$\mathbb{E}_S \left[ \mathcal{R} \left( f_{S,\lambda}^{\mathrm{KRR}} \right) \right] \leqslant \left( 1 + \frac{192}{\lambda n} \right) \mathbb{E}_S \left[ \frac{1}{n} \sum_{t=1}^n \left( f_{S,\lambda}^{\mathrm{KRR}}(\boldsymbol{x}_t) - y_t \right)^2 \right] .$$

It is easy to see[4] that this bound implies the following convergence rate

$$\mathbb{E}_S \left[ \mathcal{R} \left( f_{S,\lambda}^{\mathrm{KRR}} \right) \right] - \mathcal{R}(f_\rho) \leqslant \left( 1 + \frac{192}{\lambda n} \right) \lambda^{2\beta} \|L_K^{-\beta} f_\rho\|_\rho^2 + \frac{192 \mathcal{R}(f_\rho)}{\lambda n} .$$

This convergence rate matches only half of our bound. In particular, it does not contain the term that depends in the capacity of the RKHS through $b$. Also, the theorem in Shalev-Shwartz and Ben-David [2014] holds only for $\lambda \geqslant \frac{4}{m}$. This essentially prevents the setting of $\lambda = 0$ and the possibility to achieve the rate of $n^{-1}$ in the case that $\beta = \frac{1}{2}$ and $\mathcal{R}(f_\rho) = 0$.

Another similar bound is the Leave-One-Out analysis in Zhang [2003], which gives

$$\mathbb{E}_S \left[ \mathcal{R} \left( f_{S,\lambda}^{\mathrm{KRR}} \right) \right] \leqslant \left( 1 + \frac{2}{\lambda n} \right)^2 \min_{f \in \mathcal{H}_K} \lambda \|f\|^2 + \frac{1}{n} \sum_{t=1}^n (f(\boldsymbol{x}_t) - y_t)^2 .$$

As for the stability bound, using Theorem 4, this bound implies the following bound for $\lambda > 0$:

$$\mathbb{E}_S \left[ \mathcal{R} \left( f_{S,\lambda}^{\text{KRR}} \right) \right] - \mathcal{R}(f_\rho) \leqslant \left( 1 + \frac{2}{\lambda n} \right)^2 \lambda^{2\beta} \| L_K^{-\beta} f_\rho \|_\rho^2 + \left( \frac{4}{\lambda n} + \frac{4}{\lambda^2 n^2} \right) \mathcal{R}(f_\rho) \, .$$

Hence, this bound suffers from the same problems of the stability bound; it is suboptimal with respect to the capacity of the space and the presence of the square always makes the $\lambda$ that minimizes the risk bound bounded away from zero.

The best known results for nonparametric least square under Assumptions 1–3 are obtained by KRR [Lin et al., 2018] and by stochastic least square [Dieuleveut and Bach, 2016], with the rate

$$\mathbb{E}_S \left[ \mathcal{R} \left( f_{S,\lambda} \right) \right] - \mathcal{R}(f_\rho) \leqslant \begin{cases} O \left( n^{-\frac{2\beta}{2\beta+b}} \right), & \text{if } 2\beta + b \geqslant 1, \\ O \left( n^{-2\beta} \right), & \text{otherwise.} \end{cases}$$

This kind of rates are suboptimal in the regime $2\beta + b < 1$. In contrast, our result achieves the optimal rate in all regimes. Also, these rates do not depend in any way on the risk of the optimal function $f_\rho$. Hence, they never support the choice of a regularization parameter being zero. Pillaud-Vivien et al. [2018] call the regime $2\beta + b < 1$ the "hard" problems and prove that SGD with multiple passes achieves the optimal rate for a subset of the hard problems However, their result makes an additional assumption on the infinity norm of the functions in $\mathcal{H}_K$. Under the same assumption, Steinwart et al. [2009] present a convergence rate of $O(n^{-\frac{2\beta}{2\beta+b}})$ in all regimes for truncated KRR.

The only result we are aware of that shows an acceleration in the low noise case is Orabona [2014]. Using a SGD-like procedure that does not require to set parameters, he proves a rate of $O(n^{-\frac{2\beta}{2\beta+1}})$ that accelerates to $O(n^{-\frac{2\beta}{\beta+1}})$ when $\mathcal{R}(f_\rho) = 0$, for smooth and Lipschitz losses.

Turning to KRR used for classification, in the extreme case of the Tsybakov's noise condition (also called Massart low noise condition [Massart and Nédélec, 2006]) Yao et al. [2007] proved an exponential rate of convergence. However, this is specific to the classification case only and it does not apply to the regression setting. Under stronger assumptions, i.e. data separable with margin, the same effect was already proved in Zhang [2001]. It is also interesting to note that these results require a non-zero implicit or explicit regularization.

More recently, Hastie et al. [2019] showed[5] an asymptotic result (as $n \to \infty$) that the best regularization parameter $\lambda$ of ridge regression is $0$ when there is no label noise (i.e., $\mathcal{R}(f_\rho) = 0$) and $\beta = \frac{1}{2}$. Their result aligns well with ours, but we are not limited to asymptotic regimes nor finite dimensional spaces. On the other hand, our guarantee is an upper bound on the risk rather than an equality.

## 5 Empirical Validation

In this section, we empirically validate some of our theoretical findings. Inspired by Pillaud-Vivien et al. [2018], we consider a spline kernel of order $q \geqslant 2$ where $q$ is even [Wahba, 1990, Eq. (2.1.7)]. Specifically, we define

$$\Lambda_q(x, x') = 1 + 2 \sum_{k=1}^{\infty} \frac{\cos(2\pi k(s-t))}{(2\pi k)^q} \, .$$

and use the kernel $K(x, x') = \Lambda_{1/b}(x, x')$ for some $b \in [0,1]$. We consider the uniform distribution $\rho_{\mathbb{X}}$ on $\mathbb{X} = [0,1]$ and define the target function to be $f^\star(x) = \Lambda_{\frac{\beta}{b} + \frac{1}{2}}(x, 0)$ for $x \in \mathbb{X}$. We define the observed response of $x$ to be $f^\star(x) + B$ where $B$ is a uniform random variable $[-\epsilon, \epsilon]$. One can show that this problem satisfies Assumptions 1–3 [Pillaud-Vivien et al., 2018].

For each $n$ in fine-grained grid points in $[10^2, 10^3]$ and $\lambda$ in another fine-grained set of numbers, we draw $n$ training points, compute $f_n$ by Algorithm 1, and estimate its excess risk by a test set. Finally, for each $n$ we choose the $\lambda$ that minimizes the average excess risk. We repeat the same 5 times. First, we set $b = \frac{1}{8}$ and $\beta = \frac{7}{16}$, and $\epsilon = 0.1$. Figure 1(a) plots the excess risk of the best $\lambda$'s vs $n$, which approximately achieves the predicted rate $n^{-\frac{7}{8}}$.

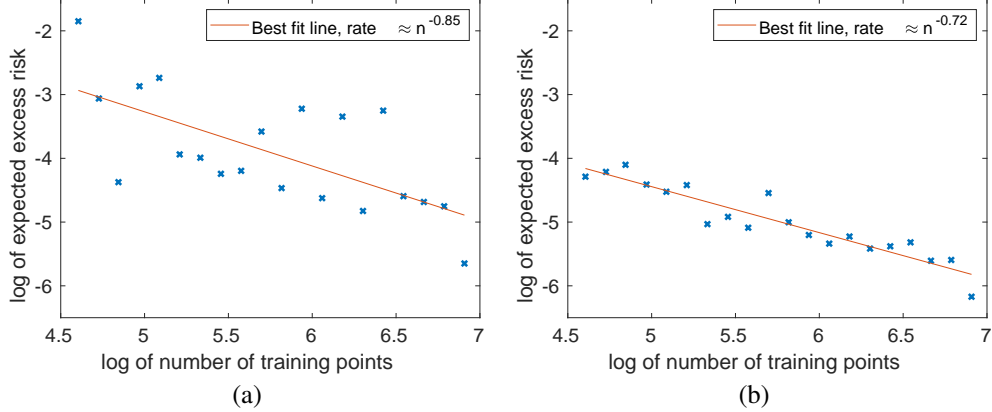

Figure 1: Expected excess risk of KTR[3] vs the number of training points on a synthetic dataset with a spline kernel. (a) and (b) show two different difficulties of the task, as parametrized by $\beta$ and $b$.

To verify our improved rate in the regime $2\beta + b < 1$, we also consider the case of $\beta = \frac{1}{4}$, $b = \frac{1}{6}$, and $\epsilon = 0.1$. Figure 1(b) plots the excess risk of the best $\lambda$'s vs $n$, which approximately achieves the predicted rate $n^{-\frac{3}{4}}$ rather than the slow rate $n^{-\frac{1}{2}}$ of prior art.[6]

## 6   Discussion and Open Problems

We have presented a new algorithm for kernel-based nonparametric least squares that achieves optimal generalization rates with respect to the source condition and complexity of the RKHS. Moreover, faster rates are possible when the Bayes risk is zero, even when the optimal predictor is not in $\mathcal{H}_K$.

One natural open problem is to prove similar guarantees for KRR. We conjecture that the randomization used in our analysis is not strictly necessary; it only greatly simplifies the proof. One may try to prove that the generalization error of KRR is nonincreasing with $n$ in which case the randomization only harms the generalization and thus implies that KRR enjoys the same error bound as KTR[3]. Such a claim is, unfortunately, not true, shown by Viering et al. [2019, Example III] where the error rate of KRR can increase with $n$.

It would also be interesting to prove lower bounds for the $\mathcal{R}(f_\rho) = 0$ case, to understand if the obtained rates are optimal or not. Furthermore, alleviating the boundedness assumption (Assumption 1) would be interesting, possibly with some mild moment conditions that appear in Hsu et al. [2012], Audibert and Catoni [2011] and Hsu and Sabato [2016].

One consequence of our work is that it shows a gap between the best-known bounds for SGD and ERM-based algorithms. Indeed, before this work, the rates of SGD and ERM-based algorithms (e.g., KRR) under Assumptions 1–3 were the same. It would be interesting to understand if some variants of SGD can achieve the optimal rates or if there is indeed a clear separation between the rates.

The limitation of this work is mainly with regards to the parametrization of the problem via the source condition and the complexity of the RKHS. Specifically, our rates are only valid for $\beta \leqslant 1/2$ (see Assumption 2), due to use of Theorem 4. However, this is unlikely to be a limitation of the analysis but rather a consequence of the use of a regularizer and the consequent "saturation" phenomenon, see discussion in Yao et al. [2007]. Another limitation of our framework is that it is well-known that the guarantee on the approximation error in Theorem 4 is non-trivial for a Gaussian kernel with fixed bandwidth only if $f_\rho \in C^\infty$ [Smale and Zhou, 2003]. While this is a strong condition from a mathematical point of view, it is unclear how strong it is for real-world problems, where the bandwidth of the Gaussian kernel is often tuned.

Finally, we believe the assumptions considered too strong in the theory community can be reconsidered with modern machine learning tasks. Indeed, most results in the community have ignored the case of $\mathcal{R}(f_\rho) = 0$, perhaps due to the fact that it was considered too strong as a condition. However, most of the visual perception tasks on which modern machine learning has been successful seem to satisfy this assumption; for example, humans have zero or very close to zero error in recognizing cats versus dogs from a photograph. In this view, a more ambitious open problem is to find the correct characterization of "easiness" for real-world problems, rather than using mathematically appealing ones.

## Acknowledgements

The authors thank Junhong Lin, Lorenzo Rosasco, and Alessandro Rudi for the comments and discussions on this work. This material is based upon work supported by the National Science Foundation under grant no. 1908111 "Collaborative Research: TRIPODS Institute for Optimization and Learning".

## Footnotes

*This work was done while the author was at Boston University.

[2]Note that the ordering of the elements in $S$ is immaterial, but our algorithm will depend on it. So we can just consider $S$ ordered according to an initial random shuffling.

[3]When $b = 0$ the space is finite dimensional, hence $\beta$ can only have value 0 or $1/2$ and there is no convergence to the Bayes risk when $\beta = 0$.

[4]For completeness, the proof is in Theorem 5 in the Appendix.

[5]To see this, set $\sigma^2 = 0$ in Hastie et al. [2019, Theorem 6].

[6]We remark that the considered kernel satisfies an extra assumption (e.g., Pillaud-Vivien et al. [2018, Assumption (A3)]) that in fact allows KRR to achieve the same optimal rate as ours. We are not aware of simple problems where that condition is not satisfied. However, our theory clearly does not make such an assumption yet achieves the optimal rate.

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
