[Supplementary Material]

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

# A  Auxiliary Results

The following result will be used in the proof of Lemma 1.

**Lemma 3.** *Let $0 \leqslant x \leqslant M$. Then, for all $b \in [0,1]$, we have*

$$\ln(1+x) \leqslant \min_{b \leqslant a \leqslant 1} \left(\frac{a}{b}\right)^a \ln^{1-a}(1+M)x^b \leqslant \min\left(\frac{1}{b}, \ln^{1-b}(1+M)\right)x^b.$$

*Proof.* For all $a$ in $[0,1]$, we have

$$\ln(1+x) = \ln^a(1+x)\ln^{1-a}(1+x) \leqslant \ln^a(1+x)\ln^{1-a}(1+M) \leqslant \left(\frac{a}{b}\right)^a \ln^{1-a}(1+M)x^b,$$

where in the last inequality we used the inequality $\ln(1+x) \leqslant \frac{1}{y}x^y, \forall y \leqslant 1$, with $y = \frac{b}{a}$.  □

We can now prove Lemma 1.

*Proof of Lemma 1.* For this proof, we need some additional notation related to learning in RKHS. Defining $K_{\boldsymbol{x}} := K(\cdot, \boldsymbol{x})$, we have the covariance operator and its empirical version

$$T_K = \int_{\mathbb{X}} \langle \cdot, K_{\boldsymbol{x}} \rangle K_{\boldsymbol{x}} d\rho(\boldsymbol{x}) \qquad \text{and} \qquad T_n = \frac{1}{n} \sum_{\boldsymbol{x}_t \in S} \langle \cdot, K_{\boldsymbol{x}_t} \rangle K_{\boldsymbol{x}_t}.$$

We have that $T_K$ is positive, self-adjoint, and trace class [Rosasco et al., 2010, Proposition 8]. Note that $T_K$ has domain and range equal to $\mathcal{H}_K$.

$T_K$ and $L_K$ are related operators, indeed they can be written as $R^\star R$ and $RR^\star$ respectively, for an appropriate operator $R$, where $R^\star$ denotes the adjoint of $R$ [Rosasco et al., 2010]. For our aims, it is enough to note that $\mathbb{E}_S[T_n] = T_K$ and $L_K$ and $T_K$ have the same non-zero eigenvalues [Rosasco et al., 2010, Proposition 8] and $T_n$ and $\frac{1}{n}K_n$ have the same non-zero eigenvalues [Rosasco et al., 2010, Proposition 9].

Hence, using the concavity of the log det with Jensen's inequality and the above observations, we have

$$\mathbb{E}_S\left[\ln \frac{|\lambda I_n + \frac{1}{n}K_n|}{|\lambda I_n|}\right] = \mathbb{E}_S\left[\ln\left|I_n + \frac{1}{\lambda n}K_n\right|\right] \leqslant \ln\left|I_n + \frac{1}{\lambda}T_K\right| = \sum_{i=1}^{\infty} \ln\left(1 + \frac{\lambda_i}{\lambda}\right)$$

$$\leqslant \min\left(\ln^{1-b}\left(1 + \frac{1}{\lambda}\right), \frac{1}{b}\right)\frac{\sum_{i=1}^{\infty}\lambda_i^b}{\lambda^b} = \min\left(\ln^{1-b}\left(1 + \frac{1}{\lambda}\right), \frac{1}{b}\right)\frac{\text{Tr}[L_K^b]}{\lambda^b},$$

where $\lambda_i$ are the eigenvalues of $L_K$ and in the second inequality we used Lemma 3.

In the case of $b = 0$, we know that there exists a finite number of nonzero eigenvalues of $L_K$ since otherwise Assumption 3 is violated. Let $d' = \text{Tr}[L_K^0]$. Then,

$$\sum_{i=1}^{\infty} \ln\left(1 + \frac{\lambda_i}{\lambda}\right) = \sum_{i=1}^{d'} \ln\left(1 + \frac{\lambda_i}{\lambda}\right) = d' \ln\left(\left(\prod_{i=1}^{d'}\left(1 + \frac{\lambda_i}{\lambda}\right)\right)^{\frac{1}{d'}}\right) \leqslant d' \ln\left(\frac{1}{d'}\sum_{i=1}^{d'}\left(1 + \frac{\lambda_i}{\lambda}\right)\right)$$

$$\leqslant d' \ln\left(1 + \frac{1}{d'\lambda}\right),$$

where in the first inequality we used the inequality of arithmetic and geometric means.  □

The following theorem elaborates our argument on $f_{S,\lambda}^{\text{KRR}}$ in Section 4.

**Theorem 5.** *Let $A > 0$ and denote by $f_{S,\lambda}$ the solution of KRR over a training set $S$ and parameter $\lambda > 0$. When Assumption 2 holds, the following inequality*

$$\mathbb{E}_S[\mathcal{R}(f_{S,\lambda})] \leqslant \left(1 + \frac{A}{\lambda n}\right)\frac{1}{n}\sum_{t=1}^{n}(f_{S,\lambda}(\boldsymbol{x}_t) - y_t)^2,$$

*implies that*

$$\mathbb{E}_S[\mathcal{R}(f_{S,\lambda})] \leqslant \left(1 + \frac{A}{\lambda n}\right)\lambda^{2\beta}\|L_K^{-\beta}f_\rho\|_\rho^2 + \frac{A\mathcal{R}(f_\rho)}{\lambda n}.$$

*Proof.* As in the proof of Theorem 1, define $f_\lambda = \text{argmin}_{f \in \mathcal{H}_K} \lambda\|f\|^2 + \mathcal{R}(f)$. Then, we have

$$
\mathbb{E}_S[\mathcal{R}(f_{S,\lambda})] \leqslant \left(1 + \frac{A}{\lambda n}\right) \mathbb{E}_S \left[\frac{1}{n}\sum_{t=1}^n (f_{S,\lambda}(\boldsymbol{x}_t) - y_t)^2\right]
$$

$$
\leqslant \left(1 + \frac{A}{\lambda n}\right) \mathbb{E}_S \left[\lambda\|f_{S,\lambda}\|^2 + \frac{1}{n}\sum_{t=1}^n (f_{S,\lambda}(\boldsymbol{x}_t) - y_t)^2\right]
$$

$$
\leqslant \left(1 + \frac{A}{\lambda n}\right) \mathbb{E}_S \left[\lambda\|f_\lambda\|^2 + \frac{1}{n}\sum_{t=1}^n (f_\lambda(\boldsymbol{x}_t) - y_t)^2\right]
$$

$$
= \left(1 + \frac{A}{\lambda n}\right) \left(\lambda\|f_\lambda\|^2 + \mathcal{R}(f_\lambda)\right)
$$

$$
= \left(1 + \frac{A}{\lambda n}\right) \left(\lambda\|f_\lambda\|^2 + \mathcal{R}(f_\lambda) - \mathcal{R}(f_\rho)\right) + \mathcal{R}(f_\rho) + \frac{A}{\lambda n}\mathcal{R}(f_\rho)
$$

$$
\leqslant \left(1 + \frac{A}{\lambda n}\right) \lambda^{2\beta}\|L_K^{-\beta} f_\rho\|_\rho^2 + \mathcal{R}(f_\rho) + \frac{A}{\lambda n}\mathcal{R}(f_\rho),
$$

where in the last inequality we used Theorem 4. Reordering the terms, we have the stated bound. $\square$

## B Proof of Theorem 2

Let us first state the motivation. Note that Theorem 1 does not work with $\lambda = 0$. However, if we assume $\lambda = 0$ works for now, then under the stated conditions of this corollary, the excess risk bound of Theorem 1 is $\lambda\|L_K^{-1/2} f_\rho\|_\rho^2 + \frac{1}{n}\|L_K^{-1/2} f_\rho\|_\rho^2$. This implies that $\lambda = 0$ is the optimal choice, resulting in the desired bound

$$
\frac{1}{n}\|L_K^{-1/2} f_\rho\|_\rho^2 = \frac{1}{n}\|f_\rho\|^2 .
$$

Since one cannot set $\lambda = 0$, the goal here is, therefore, to show rigorously that the desired bound above is achieved when $\lambda$ is exactly 0.

We will use the following definitions. Define the sampling operator $R_n : \mathcal{H}_K \to \mathbb{R}^n$ as $R_n f = [f(\boldsymbol{x}_1), \dots, f(\boldsymbol{x}_n)]$, where $\boldsymbol{x}_1, \dots, \boldsymbol{x}_n \in S$. The adjoint operator $R_n^* : \mathbb{R}^n \to \mathcal{H}_K$ is $R_n^* \boldsymbol{w} = \sum_{i=1}^n w_i K(\boldsymbol{x}_i, \cdot)$. Hence, we have

$$
\langle R_n^* \boldsymbol{w}, f \rangle = \sum_{i=1}^n w_i f(\boldsymbol{x}_i),
$$

where $\boldsymbol{w} \in \mathbb{R}^n$. This allows to redefine the Gram matrix over the $n$ samples of $S$ as $K_n = R_n R_n^*$ [Rosasco et al., 2010].

Now, the assumption that $\mathcal{R}(f_\rho) = 0$ implies that $y_i = f_\rho(\boldsymbol{x}_i)$, $\rho_X$-almost surely. Hence, we can write $R f_\rho = \boldsymbol{y} = [y_1, \dots, y_n]^\top$, the vector of labels in the training set $S$. So, $\rho_X$-almost surely, we can write that

$$
\boldsymbol{y}^\top K_n^{-1} \boldsymbol{y} = \langle R_n^* (R_n R_n^*)^{-1} R_n f_\rho, f_\rho \rangle,
$$

where we used the fact that $\beta = \frac{1}{2}$.

We claim that

$$
\mathbb{E}_S[\boldsymbol{y}^\top K_n^{-1} \boldsymbol{y}] \leqslant \|f_\rho\|^2 = \|L_K^{-1/2} f_\rho\|_\rho^2 . \tag{1}
$$

From the assumption that $K_n$ is full rank, we have that the operator $Q = R_n^* (R_n R_n^*)^{-1} R_n$ is a projection operator, because $Q^2 = Q$, and its eigenvalues are 0 or 1. This implies that $\langle R_n^* (R_n R_n^*)^{-1} R_n f_\rho, f_\rho \rangle \leqslant \|f_\rho\|^2$, $\rho_X$-almost surely, which proves the claim.

Let us use the notation $\ell_t$ and $\ell_t'$ defined in the proof of Theorem 1. By Zhdanov and Kalnishkan [2013, Corollary 1], we have the following well defined for $\lambda \neq 0$:

$$
\frac{1}{n}\sum_{t=1}^n \frac{\ell_t}{1 + d_t/(n\lambda)} = \min_{f \in \mathcal{H}_K} \frac{1}{n}\sum_{t=1}^n (f(\boldsymbol{x}_t) - y_t)^2 + \lambda\|f\|^2 = \lambda \boldsymbol{y}^\top (K_n + n\lambda I)^{-1} \boldsymbol{y} .
$$

Define $A_\lambda = \boldsymbol{y}^\top (K_n + n\lambda I)^{-1}\boldsymbol{y}$. We claim that

$$\sum_{t=1}^{n} \frac{\ell_t}{n\lambda + d_t} = A_\lambda \quad \text{for} \quad \lambda = 0 .\tag{2}$$

To show this, it suffices to show that $\lim_{\lambda\to 0}\sum_{t=1}^{n}\frac{\ell_t}{n\lambda+d_t} = \lim_{\lambda\to 0} A_\lambda$ since both sides are continuous at $\lambda = 0$. Then,

$$1 = \frac{\lim_{\lambda\to 0}\sum_t \frac{\ell_t}{1+d_t/(n\lambda)}}{\lim_{\lambda\to 0} n\lambda A_\lambda} \overset{(a)}{=} \lim_{\lambda\to 0}\frac{\sum_t \frac{\ell_t}{1+d_t/(n\lambda)}}{n\lambda A_\lambda} = \lim_{\lambda\to 0}\frac{\sum_t \frac{\ell_t}{n\lambda+d_t}}{A_\lambda} \overset{(b)}{=} \frac{\lim_{\lambda\to 0}\sum_t \frac{\ell_t}{n\lambda+d_t}}{\lim_{\lambda\to 0} A_\lambda} .$$

where both $(a)$ and $(b)$ is due to the fact that the existence of $\lim_{x\to 0} g(x)$ and $\lim_{x\to 0} h(x)$ implies $\lim_{x\to 0}\frac{g(x)}{h(x)} = \frac{\lim_{x\to 0} g(x)}{\lim_{x\to 0} h(x)}$. This proves the claim.

Now, when $\lambda = 0$, we have the following bound on the online average loss:

$$\frac{1}{n}\sum_{t=1}^{n} \ell'_t \leqslant \frac{1}{n}\sum_{t=1}^{n} \ell_t = \frac{1}{n}\sum_{t=1}^{n} \ell_t \cdot \frac{d_t}{d_t} \overset{(a)}{\leqslant} \frac{1}{n}\sum_{t=1}^{n} \frac{\ell_t}{d_t} \overset{(2)}{=} \frac{1}{n} A_0 .$$

where $(a)$ is by $d_t \leqslant 1$. This implies that

$$\mathbb{E}[\mathcal{R}(f_{S,0})] = \mathbb{E}\left[\frac{1}{n}\sum_{t=1}^{n} \ell'_t\right] \leqslant \frac{1}{n}\mathbb{E}[A_0] \overset{(1)}{\leqslant} \frac{1}{n}\|f_\rho\|^2 .$$

## C  Proof of Corollary 1

The proof is based on the risk bound of Theorem 1 that is minimum of multiple bounds, each of which achieving the minimum depending on the given problem parameters.

First, we have the risk bounded by

$$\lambda^{2\beta}\|L_K^{-\beta} f_\rho\|_\rho^2 + \frac{\lambda^{2\beta-1}\|L_K^{-\beta} f_\rho\|_\rho^2}{n} + \frac{\mathcal{R}(f_\rho)}{\lambda n} .\tag{3}$$

Note that $\lambda$ has to be decreasing in $n$ since otherwise the first term remains constant. This means that the second term is dominated by the third term for a large enough $n$. Thus, it remains to find $\lambda$ that minimizes $\lambda^{2\beta}\|L_K^{-\beta} f_\rho\|_\rho^2 + \frac{\mathcal{R}(f_\rho)}{\lambda n}$ using elementary algebra. The minimum is

$$\Theta\left(\left(\frac{\mathcal{R}(f_\rho)}{n}\right)^{\frac{2\beta}{2\beta+1}} \cdot (\|L_K^{-\beta} f_\rho\|_\rho^2)^{\frac{1}{2\beta+1}}\right) \quad \text{with} \quad \lambda = \left(\frac{\frac{1}{n}\mathcal{R}(f_\rho)}{2\beta\|L_K^{-\beta} f_\rho\|_\rho^2}\right)^{\frac{1}{2\beta+1}} .$$

Second, in the case where $\mathcal{R}(f_\rho) = 0$, the third term of (3) disappears. Thus, it remains to find $\lambda$ that minimizes $\lambda^{2\beta}\|L_K^{-\beta} f_\rho\|_\rho^2 + \frac{\lambda^{2\beta-1}\|L_K^{-\beta} f_\rho\|_\rho^2}{n}$. The minimum is

$$\Theta\left(\|L_K^{-\beta} f_\rho\|_\rho^2 \cdot n^{-2\beta}\right) \quad \text{with} \quad \lambda = \frac{1}{n}\left(\frac{1}{2\beta} - 1\right) .$$

Note that the case of $\beta = 1/2$ here cannot be derived by Theorem 1 since the optimal $\lambda$ is exactly 0. This case is instead supported by Theorem 2.

Third, we have another risk bound of

$$\lambda^{2\beta}\|L_K^{-\beta} f_\rho\|_\rho^2 + \frac{4Y^2 \operatorname{Tr}[L_K^b]}{\lambda^b n \cdot b} .$$

The minimum is

$$\Theta\left(\left(\frac{Y^2 \operatorname{Tr}[L_K^b]}{nb}\right)^{\frac{2\beta}{2\beta+b}} (\|L_K^{-\beta} f_\rho\|_\rho^2)^{\frac{b}{2\beta+b}}\right) \quad \text{with} \quad \lambda = \left(\frac{Y^2 \operatorname{Tr}[L_K^b]}{\beta n\|L_K^{-\beta} f_\rho\|_\rho^2}\right)^{\frac{1}{2\beta+b}} ,$$

where we remark that it is possible to remove $(1/b)^{\frac{2\beta}{2\beta+b}}$ (which can very large when $b$ is small) in the minimum at the price of a logarithmic factor using another risk bound of $\lambda^{2\beta}\|L_K^{-\beta}f_\rho\|_\rho^2 + \frac{4Y^2\operatorname{Tr}[L_K^b]}{\lambda^b n}\ln^{1-b}\left(1+\frac{1}{\lambda}\right)$.

For the case of $b=0$ and $\beta=1/2$, one can derive a tighter risk bound from the proof of Theorem 1 by invoking the second part of Lemma 1 instead of the first part:

$$\lambda\|L_K^{-1/2}f_\rho\|_\rho^2 + \frac{4Y^2}{n}\cdot\ln\left(1+\frac{1}{\operatorname{Tr}[L_K^0]\lambda}\right)\operatorname{Tr}[L_K^0].$$

Tuning $\lambda = \frac{1}{n\|L_K^{-1/2}f_\rho\|_\rho^2}$, we get a bound $\Theta(n^{-1}\operatorname{Tr}[L_K^0]\ln(1+\frac{n}{\operatorname{Tr}[L_K^0]}))$.

Clearly, given the problem parameters $\beta$, $b$, $Y$, $n$, $\operatorname{Tr}[L_K^b]$, and $\|L_K^{-\beta}f_\rho\|_\rho$, one can choose $\lambda$ that leads to the smallest risk bound among all of the above, which concludes the proof.