[Reviews · NeurIPS 2019]

Reviewer 1



1) Summary of the paper The paper introduces a randomized version of the truncated kernel ridge regression algorithm. The algorithm essentially selects a random subset of training points and learns a (truncated) kernel ridge regression function on the selected subset. Under certain characteristic assumptions on the complexity of the function class in which the optimal function lies and on the complexity of the RKHS, the paper shows that the algorithm achieves optimal generalization guarantees. This is an improvement over the existing results in this setting in one of the regimes of the problem space. Additionally, the authors show that under a zero Bayes risk condition, the algorithm achieves a faster convergence rate to the Bayes risk. The main contribution of the paper lies in adapting the proof techniques used in the online kernel regression literature to the standard kernel regression setting. 2) Main comments The structure and writing of the paper are fair and easy to follow, and the results are put into perspective with respect to the current literature. The paper also provides clear mathematical intuition behind the various assumptions made for the analysis. It is also fair to argue that the assumptions are standard in the kernel regression literature. However, the usability of the results requires that the assumptions are relatable to some practical setting. This is not provided/discussed in the paper. For instance, it is not clear how the assumption on the eigenvalue decay of the integral operator translates to conditions on the underlying distribution and the kernel function thereby limiting the usability of the results. On a similar note, the regime under which the algorithm improves over existing results needs further discussion to emphasize the relevance of the obtained results. One of the motivations of the paper are to address the discrepancy of the role of regularization in theory vs practice. The authors address this issue under the setting of zero / near zero Bayes risk condition. The authors briefly justify the usability of this setting through a visual classification example. However, unlike in the classification setting, this seems to be a very strong assumption in the regression setting (a distinction the authors clearly make in the discussion of their theoretical findings). Furthermore, the main contribution of the paper stems from the adaptation of the techniques used in the online kernel regression literature to the kernel regression setting. A further discussion and emphasis on ways to adapt the techniques to more general settings in kernel regression could have further improved the usability of the paper. 3) Minor comments and typos l. 36: missing ' after "easiness" Algorithm 1 is often referred to as Algorithm 3 in the paper. Algorithm 1: if I understand correctly, the sampling is done after solving n minimization problems. Why not first sample k, and only compute f_k? l. 227: missing space after "Figure 5" 4) After rebuttal and discussion The rebuttal adequately addressed my concerns, and I am increasing my score. I think the paper should be accepted.

Reviewer 2



The derived results for KTR3 with non-zero regularization parameters is a strict improvement in the regime $2\beta + b<1$ upon the best-known bound of KRR and stochastic gradient methods. The derived results for KTR3 also indicate that in the regime of easy problems the best among of regularization is exactly zero. -------------After rebuttal----------------- The comparisons with related results have been made clearly by Fischer/Steinwart in Table 1 of the nice paper (https://arxiv.org/pdf/1702.07254.pdf). (For ease of comparisons, I use the same notations from that Table in the following). The focus of the paper is on the case $\alpha =1$. In my opinion, this paper gave optimal rates for the case $\alpha =1$, and the results indeed improve previous results, closing a gap (but only for the case $\alpha =1$). Note that for the general case $0<\alpha\leq 1$, it is still an unresolved problem. And in my opinion, the technique of the paper under review may demonstrate some further new results for the general case $\alpha\leq 1$. Thus, I keep my scores unchanged. A further comment on the nice paper (https://arxiv.org/pdf/1702.07254.pdf) is needed in the final version. The derived results are very interesting, closing a long-standing gap between upper and lower bounds. The proof technique developed may be also interesting to the community. The paper is well written. I did a high-level check of the proof. I believe the proof should be correct. Minor comments:\Line 70 on Page 2, $X$ is compact..\Line 71 on Page 2, a citation is needed here for $f_{\rho} \in \bar{H_K}$; ... which is the closer of $H_K$ in $L_{\rho}^2$ \Line 80 on page 3, $L_K$ is compact ?\

Reviewer 3



*this paper is nicely written and the results are presented in a rigorous and clean way, I like that paper! *also, the results are important since it was not clear to the community if these improved bounds also hold in that regime *however I would like to ask the authors for being more clear: in l. 38/39 they write that these bounds are mini-max optimal; in fact, in that regime there are no known lower bounds! so claiming optimality is a bit too much here! please be more careful, also on p.4, l.127-131 (the regime of Caponnetto and DeVito is different) *I do not fully get the effect of randomization. Can you prove your results without randomization (and usingthe full dataset)? Your randomization-strategy allows to build an estimator with less data then the full dataset. My intuition is that this should have an empirical effect. Surely, these rates cannot be established when using only a subset of the data (without additional assumptions). How much do you "loose" when implementing your algorithm? In your experiments you choose the full sample. This deserves a more detailed discussion.

[Author Response · NeurIPS 2019]

**To Reviewer 1:**

We thank the reviewer for providing constructive feedback and suggestions. Also, in the final version we will fix the typos and make the minor improvements the reviewer suggests.

**On the assumptions.** As you also noted, the source condition and the eigenvalue decay are fairly standard in the nonparametric regression setting. By now, the number of papers (and books!) using these two parametrizations is quite large. So, for example, the eigenvalue decay of the Gaussian Kernel is well-known (see the very recent results in Belkin, COLT'18 and references therein). In particular, as briefly explained in Section 2, the source condition and the eigenvalue decay parametrize the difficulty of the problem and the "finiteness" of the space. Hence, our rate improves the known rates for "difficult" problems with low effective dimension. However, given that our result is based on a minor modification of regularized least square, it is likely that the previous analyses were loose, not that our algorithm is intrinsically better! In this view, the specific regime in which our rates are better is not really important. On the other hand, in our opinion, closing the gap between upper and lower bounds and pointing out possible major problems in previous work through a completely novel analysis are major contributions.

**On "strong" assumptions.** Our final comments on the bias of the community towards "weak assumptions" was exactly to provoke a discussion in this sense and a better judgment on these issues, rather than justifying our results. So, we are happy that the reviewer engaged with us in this discussion! In this view, we abstain from judging how "strong" is the case of zero Bayes error w.r.t. the square loss: It is completely a problem-dependent judgment rather than a universal one. Instead, we just consider it an interesting setting that researchers have ignored for a long time. Moreover, we do plan to extend the results we presented to smooth classification losses, as the squared hinge loss. In that setting, the same results are expected through an Online-Newton-Step analysis. Indeed, the work in Orabona (2014) already shows an acceleration for zero Bayes error for any smooth and Lipschitz loss, (even classification ones like the smoothed hinge loss), but the acceleration appears inferior to the one we can show for the square loss. So, we believe this is an interesting area to explore.

**To Reviewer 2:**

We thank the reviewer for providing constructive feedback. We will improve accordingly in the final version.

**To Reviewer 3:**

We thank the reviewer for raising interesting questions and suggestions.

**On the lower bound.** We actually believe that the lower bound is known and matching our upper bound. However, we did use the wrong citation, thanks for pointing it out! In particular, the lower bound is widely discussed in Section 4.2 of Pillaud-Vivien et al. (2018), that in turn is based on the theorems in

S. Fischer and I. Steinwart. Sobolev norm learning rates for regularized least-squares algorithm. Fakultät für Mathematik und Physik, Universität Stuttgart, 2017.

We will make it clear in the final version.

**On the experiments.** We are *not* doing the full sample version, and we do plot the *expected* value of the risk, as written on the $y$ axis. So, we exactly compute the expectation with respect to the randomization of the algorithm using $k$ from 0 to $n-1$, while we estimate the test error for each $k$ using a finite test set. We will make this clear in the final version.

**On the effect of randomization.** We have not performed a thorough empirical comparison with the full sample version, and we are not sure of the exact effect of randomization (besides the fact that it gives us a way to obtain a good theoretical bound). You have raised an interesting question, which would be an interesting future work. We believe this is quite nontrivial for the following reason: If the error rate of the kernel ridge regression is monotonic with the data size, then the randomization would necessarily harm the prediction error. However, in general it turns out the error rate can be non-monotonic by a recent study by Viering et al., Open Problem: Monotonicity of Learning, COLT, 2019 (see example III therein). Specifically, the error rate of ridge regression can even *increase* with the data size in some regime.

[Meta-Review · NeurIPS 2019]

After a careful discussion among the reviewers, there is a clear consensus that the paper provides a solid contribution to the community. As a result, I would recommend acceptance for publication at NeurIPS2019. Many congratulations. One important concern that came up during the discussion is that it is unclear under which regime the paper is focusing on. As a result, it becomes difficult for the reviewers and readers to assess the actual contribution. For example, the authors need to clarify that the paper needs \beta \geq 1/2 to hold and that it considers *only* the case $\alpha=1$. To clarify this in the camera-ready version, I urge the authors to give a pointer to Remark 3.3 on p. 7 of Fischer/Steinwart https://arxiv.org/pdf/1702.07254.pdf